


# DNS (v1.0): An open source ray-tracing tool for space geodetic techniques

Florian Zus[1], Kyriakos Balidakis[1], Ali Hasan Dogan[2], Rohith Thundathil[1,3], Galina Dick[1], Jens Wickert[1,3]

[1]GFZ German Research Centre for Geosciences, Telegrafenberg 14473, Potsdam, Germany
[2]Department of Geomatics Engineering, Gaziosmanpasa University, 60150, Tokat, Türkiye
[3]Technische Universität Berlin, 10623, Berlin, Germany

*Correspondence to*: Florian Zus (zusflo@gfz-potsdam.de)

**Abstract.** We have developed an open source ray-tracing tool for space geodetic techniques. The software uses the

geometric optics approximation to calculate the signal travel time delay induced by the atmosphere between two given points. The software is written in Fortran and uses OpenMP to speed up computation. The input to the ray-tracing tool are 3D pressure-, temperature- and humidity fields. The Earths magnetic- and electron density field are optional. For the neutral atmosphere (troposphere) the software accepts the NetCDF files from the atmospheric reanalysis ERA5 and the mesoscale model WRF. For the ionosphere the software accepts electron density fields derived from IRI and Ne-quick. We review the

current status of the software and test its performance. For example, the one-to-one comparison with the open source software RADIATE shows the high speed and precision of our ray-tracing tool. We also show how our tool can be used to study higher order ionospheric effects (L-band frequencies). The two outstanding features of the ray-tracing tool compared to previous model developments, i.e., the ability to handle both the troposphere and the ionosphere and do so efficiently, make it perfectly suited for geoscientific applications.

## 1 Introduction

Atmospheric signal propagation effects are one of the largest error sources in the data analysis of space geodetic techniques (Very Long Baseline Interferometry (VLBI), Satellite Laser Ranging (SLR),  Global Navigation Satellite Systems (GNSS)

and Doppler Orbitography and Radio-positioning Integrated by Satellite (DORIS)). These effects are caused by the neutral atmosphere, hereinafter simply called the troposphere, and the ionosphere. Inaccuracies in signal propagation modelling map into errors in positioning, navigation and timing (e.g., Rothacher et al., 1998), Earth rotation parameter determination (e.g., Nilsson et al., 2017), terrestrial and celestial reference frames establishment and maintenance (e.g., MacMillan and Ma, 1997), as well as atmospheric and climate monitoring (e.g., Bock et al., 2019). In the space geodetic data analysis the signal

propagation effects for the troposphere and ionosphere are treated separately. However, the models for both effects are based





on the same key algorithm; ray-tracing. This is the algorithm that determines the (bent) signal path between the transmitter and the receiver and therefore allows the accurate calculation of the tropospheric and ionospheric delay.

In the analysis of space geodetic data the tropospheric delay is typically not calculated for each measurement but a
parameterized version of the tropospheric delay is developed beforehand. This closed form expression of the tropospheric delay is plugged into the observation equation and some tropospheric parameters are estimated (adjusted) together with the geodetic parameters such as coordinates, clocks among others. The quality of the geodetic parameter estimates depends on the quality of the tropospheric delay which depends on the accuracy of the underlying model for the troposphere and the ray-tracing algorithm. The most advanced approaches utilize data from Numerical Weather Models (NWMs) as an input for
optical (Hully et al., 2007; Bosities et al., 2020; Drożdżewski et al., 2024) and microwave frequencies (Boehm et al., 2006 ; Landskron et al., 2018a; Landskron et al., 2018b). Several ray-tracing algorithms have been developed starting from the 1960's. In the early days the ray-tracing algorithms were based on simplifying assumptions, i.e., the assumption of a spherically layered troposphere. In this case a single pressure, temperature and humidity profile is utilized to calculate the tropospheric delays. Later, ray-tracing algorithms were developed to cope with the 3D pressure-, temperature- and humidity
fields. Nafisi et al., 2012a presents a summary of the most commonly used algorithms. Setting up a ray-tracing system is not a trivial task and as of this writing very few resources are publicly available, namely the software package RADIATE (Hofmeister et al., 2016) and the software package SPD (Petrov, 2015). Utilizing NWM data and not the data from a climatology means that ray-tracing must be performed on a routine basis and thus a ray-tracing algorithm that is both accurate and fast is beneficial.


With regards to the ionospheric delay the treatment in the analysis of space geodetic data is somewhat different. The reason is that the ionosphere is a dispersive medium so that measurements taken with different frequencies and the appropriate combination of these measurements allows one to remove signal propagation effects caused by the ionosphere to a large extent. The remaining signal propagation effects caused by the ionosphere, the so-called higher-order ionospheric effects,
can be neglected in many applications. However, in precise applications they must be taken into account (Petrie et al., 2011). These higher-order ionospheric effects are caused by ray-path bending effects among others. Hence a ray-tracing algorithm is required too. Typically, the higher-order ionospheric effects are calculated using idealized electron density fields. For example, Hoque and Jakowski, 2008 utilized idealized electron density profiles at predefined ionospheric pierce points. They also derived a parameterization for the higher-order ionospheric effects. Kashcheyev et al., 2012 argued that this
oversimplifies the task and hence they utilized realistic electron density fields. This means that there is no need to define some idealized electron density profile at some predefined ionospheric pierce point. Instead the electron density field is utilized as is. The prize they paid was computational complexity so that they were forced to analyse a limited geographical area and time period. They did not provide a parametrization for the higher-order ionospheric effects.





In this work we present the open source ray-tracing tool DNS ('Direct Numerical Simulation of signal travel time delay'). We demonstrate the two outstanding features of this tool compared to previous model developments: it can handle both the troposphere and the ionosphere and it does so with high speed and precision. This makes the tool perfectly suited for geoscientific applications. In Section 2 we introduce the method and data. In essence, we introduce the tropospheric delay, the ionospheric delay, the input data that can be utilized in the ray-tracing tool and technical details of the implementation. In

Section 3 we perform numerical experiments. At first, we provide the one-to-one comparisons for the calculated tropospheric delays before we provide an application example, namely the comparison with GNSS tropospheric estimates. In a similar fashion, we provide comparisons for the calculated ionospheric delays before we provide an application example, namely we study the impact of higher order ionospheric corrections in precise point positioning. The conclusion is given in Section 4.

**2 Data and Methods**

**2.1 Tropospheric Delay**

The tropospheric delay T between two points P and Q is calculated according to


$$T = \int_P^Q n \, ds - g \tag{1}$$

where $n$ stands for the index of refraction, $s$ denotes the arclength of the ray path and $g$ denotes the geometric distance between the two points. The index of refraction is related to the refractivity $N$ through


$$n = 1 + 10^{-6} N \tag{2}$$

The refractivity is a function of the pressure, temperature and humidity. For microwave frequencies we take the formula by Thayer, 1974 and for optical frequencies we take the formula by Mendes and Pavlis, 2004. The ray path follows from

Fermat's principle. In essence, the variation of the integral must vanish

$$\delta \int_P^Q n \, ds = 0 \tag{3}$$

We ignore out-of-plane bending which means that we assume that the ray path remains in the plane, which is defined by the

point P, the centre of the osculating sphere (we approximate the Earth as a sphere) and the point Q. Let [x, z(x)] denote the ray-path, then from variational calculus it follows that the ray path between the two points is obtained by solving the following differential equation (Zus et al., 2014)





$$\frac{d^2z}{dx^2} - \frac{1}{n}\left[\frac{\partial n}{\partial z} - \frac{\partial n}{\partial x}\frac{dz}{dx}\right] \cdot \left[1 + \left(\frac{dz}{dx}\right)^2\right] = 0 \tag{4}$$


Given the position of point P and Q the two point Boundary Value Problem (BVP) is solved by a finite difference scheme (Zus et al., 2014). The point P can be any point close to Earth's surface (ground-based station) and the point Q can be any point between a Low Earth Orbit (LEO) and High Earth Orbit (HEO) satellite. The point P is provided by the user as an input. The point Q is by default a Medium Earth Orbit (MEO) satellite. We also allow for a special case in which the point Q
is an extragalactic radio source (quasar) as it is the case for VLBI. For details the reader is referred to Zus et al., 2014.

It is convenient to express the tropospheric delay as a function of the elevation angle $e$ and azimuth angle $a$ in the local horizon system of the ground-based station

$T = T(e, a) \tag{5}$

For any ground-based station we calculate 120 tropospheric delays and write them to disk ('SLA' file). The spacing in azimuth is chosen to be 30° and the elevation angles are chosen to be 3°, 5°, 7°, 10°, 15°, 20°, 30°, 50°, 70°, 90°. It is up to the user to change the default selection of elevation and azimuth angles. The output files also contain the slant hydrostatic
delay SHD and the slant wet delay SWD calculated through

$$SHD = 10^{-6}\int_P^Q N_h \, ds + \int_P^Q ds - g \tag{6}$$

$$SWD = 10^{-6}\int_P^Q N_w \, ds$$


where $N_h$ denotes the hydrostatic refractivity and $N_w$ denotes the wet refractivity. In the zenith direction the SHD and SWD are called Zenith Hydrostatic Dealy (ZHD) and Zenith Wet Delay (ZWD) respectively. The separation into the hydrostatic and wet delay is useful for those interested in the parameterization of tropospheric delays, i.e., the derivation of mapping functions (e.g., Dogan et al., 2024). In addition, we calculate the difference between the ZHD from the numerical integration
and the ZHD from the empirical formula depending on the pressure at the station and write it to disk ('ZHD' file). For microwave frequencies we take the formula by Davis et al., 1985 and for optical frequencies we take the formula by Mendes and Pavlis, 2004. This file is utilized for diagnostic purposes. The ZHD differences should be on a mm level (e.g., Fan et al., 2023). Larger ZHD differences point to a problem in the underlying weather model field, the way refractivity is interpolated (extrapolated) or the numerical integration.






## 2.2 Ionospheric delay

Regarding signal propagation in the ionosphere we must distinguish between the phase and group velocity. We are only going to analyse the phase velocity. Simple modifications allow the user to analyse the group velocity. Hence, the

ionospheric delay $I_i$ between the two points P and Q is calculated according to

$$I_i = \int_P^Q n \, ds - \int_P^Q n_i \, ds_i \tag{7}$$

Here $n_i$ stands for the refractive index experienced by the phase of the signal, $s_i$ denotes the arclength of the ray path and the

index $i$ refers to the carrier frequency of the signal. The refractive index experienced by the phase of the signal is given by

$$n_i = 1 + 10^{-6} N - \frac{q}{f_i^2} E - \frac{p}{f_i^3} E \, B \cos(\phi) \tag{8}$$

where $f_i$ denotes the carrier frequency of the signal, $E$ denotes the electron density $\theta$ denotes the angle between the Earth's

magnetic field vector $B$ and the ray-tangent vector and $q$ and $p$ denote constants (Hoque et al., 2008). The variation of the integral must vanish

$$\delta \int_P^Q n_i \, ds_i = 0 \tag{9}$$

We neglect out-of-plane bending. From variational calculus, it follows that the ray path [x, z(x)] between the two points is obtained by solving the following differential equation (Zus et al., 2017)

$$\frac{d^2 z}{dx^2} - \frac{1}{\eta_i}\left[\frac{\partial \eta_i}{\partial z} - \frac{\partial \eta_i}{\partial x}\frac{dz}{dx}\right] \cdot \left[1 + \left(\frac{dz}{dx}\right)^2\right] - \frac{p}{\eta_i f_i^3}\left[\frac{\partial (E \cdot B_x)}{\partial z} - \frac{\partial (E \cdot B_z)}{\partial x}\right] \cdot \left[1 + \left(\frac{dz}{dx}\right)^2\right]^{\frac{3}{2}} = 0 \tag{10}$$

where

$$\eta_i = 1 + 10^{-6} N - \frac{q}{f_i^2} E \tag{11}$$

and $B_x$ and $B_y$ denote the x and z coordinate of the projection of Earth's magnetic field vector $B$ into the plane of constant

azimuth. The curvature term due to Earth's magnetic field is small and can be neglected. Therefore, the ray path between the two points is obtained by solving the following differential equation (Zus et al., 2017)





$$\frac{d^2 z}{dx^2} - \frac{1}{\eta_i}\left[\frac{\partial \eta_i}{\partial z} - \frac{\partial \eta_i}{\partial x}\frac{dz}{dx}\right] \cdot \left[1 + \left(\frac{dz}{dx}\right)^2\right] = 0 \tag{12}$$


The two point BVP is efficiently solved by a finite difference scheme (Zus et al., 2017).

With the definition of the tropospheric delay $T$ and the definition of the ionospheric delay $I_i$ the optical path length $O_i$ that enters the so-called observation equation be written in the familiar form


$$O_i = g + T - I_i \tag{13}$$

In the analysis of space geodetic data the effect of the ionosphere is mitigated by the dual-frequency linear combination. The residual


$$\Delta I = \frac{f_1^2}{(f_1^2 - f_2^2)}I_1 - \frac{f_2^2}{(f_1^2 - f_2^2)}I_2 \tag{14}$$

is called higher-order ionospheric correction. It is convenient to express the higher-order ionospheric correction as a function of the elevation and azimuth angel in the local horizon system of the ground-based station


$$\Delta I = \Delta I(e, a) \tag{15}$$

For any ground-based station we calculate by default 120 higher-order ionospheric corrections. The spacing in azimuth is chosen to be 30° and the elevation angles are chosen to be 3°, 5°, 7°, 10°, 15°, 20°, 30°, 50°, 70°, 90°. By default the higher-185 order ionospheric corrections are computed for the GPS L-band frequencies L1 and L2. It is up to the user to change the default selection of frequencies. We write the calculated higher-order ionospheric corrections to disk ('HOC' file). In addition, we perform a polynomial expansion (we make use of Zernike polynomials) and write the coefficients of the polynomial expansion to disk ('HOI' file). This allows a quick access to higher-order ionospheric correction for any elevation and azimuth angle. For diagnostic purposes we calculate the Vertical Total Electron Content (VTEC) at each 190 ground-based station and write it to disk ('TEC' file). The VTEC is obtained by numerical integration of the electron density in zenith direction.





We note that higher order ionospheric corrections are not sensitive to the refractivity of the neutral atmosphere. Therefore as long as one is only interested in higher order ionospheric corrections there is no need to know the state of the neutral atmosphere. Specifically, $N = 0$ implies $ds = dg$ and the higher order ionospheric corrections can be written as

$$\Delta I = \alpha + \beta + \gamma \tag{16}$$

where the three terms read as

$$\alpha = \frac{p}{f_1^2 - f_2^2}\left(\frac{1}{f_1}\int_P^Q E\,B\,cos(\phi)\,ds_1 - \frac{1}{f_2}\int_P^Q E\,B\,cos(\phi)\,ds_2\right) \tag{17}$$

$$\beta = \frac{q}{f_1^2 - f_2^2}\left(\int_P^Q E\,ds_1 - \int_P^Q E\,ds_2\right)$$

$$\gamma = g - \left(\frac{f_1^2}{f_1^2 - f_2^2}\int_P^Q ds_1 - \frac{f_2^2}{f_1^2 - f_2^2}\int_P^Q ds_2\right)$$

and agree with those provided by Kashcheyev et al., 2012. The third term $\gamma$ is caused by the ray path bending, the second term $\beta$ is due to the fact that the ray path bending is different for different carrier frequencies, and the first term $\alpha$ is caused by the Earth's magnetic field. Hereinafter we will call the sum of the second and third term ionospheric ray-path bending corrections.

**2.3 Data from Numerical (Space-) Weather Models**

The input to the ray-tracing tool are 3D pressure-, temperature- and humidity fields. Earths magnetic- and electron density field are optional. With regards to the troposphere the following options are possible:

(1) Data from the atmospheric reanalysis ERA5. The NetCDF files are available from the European Centre of Medium-Range Weather Forecasts (ECMWF) (https://www.ecmwf.int/en/forecasts/dataset/ecmwf-reanalysis-v5/). The weather models fields are available with a horizontal resolution of 0.25° × 0.25° on 37 pressure levels.

(2) Data from the operational analysis of the ECMWF. The ascii files are available from the VieVS raytracer (https://vmf.geo.tuwien.ac.at/GRIB_TXT/). The weather model fields are available with a horizontal resolution of 1° × 1° on 25 pressure levels.





(3) The output (NetCDF files) from the Weather Research and Forecasting (WRF) model (https://www2.mmm.ucar.edu/wrf/users/). The WRF model is a state of the art mesoscale numerical weather prediction system designed for both atmospheric research and operational forecasting. The user must configure the weather model and therefore specifies the horizontal and vertical resolution by himself. Currently the ray-tracing tool requires data on the cylindrical equidistant map projection.


(4) Data from the benchmark comparison campaign (https://vmf.geo.tuwien.ac.at/BMC/) (Nafisi et al., 2012a). The ascii files are provided with the software. The weather model fields are available for a station in Germany (Wettzell) and a station in Japan (Tsukuba) with a horizontal resolution of $0.1° \times 0.1°$ on 25 pressure levels.

With regards to the ionosphere the following options are possible:

(1) Data derived from the IRI-2016 (https://irimodel.org) (Bilitza, 2001). We extract electron density profiles (ranging from 80-2000km) and assemble the electron density profiles to a 3D electron density field with an horizontal resolution of $2° \times 2°$. We store the electron density fields and we make them available upon request.


(2) Data derived from the Ne-Quick2 (https://t-ict4d.ictp.it/nequick2) (Nava et al., 2008). We extract electron density profiles (ranging from 80-10000km) and assemble the electron density profiles to a 3D electron density field with an horizontal resolution of $2° \times 2°$.

(3) Data from the coupled Whole Atmosphere Model-Ionosphere Plasmasphere Electrodynamics (WAM-IPE) Forecast System (WFS). The NetCDF files are available from the NOAA Space Weather Prediction Center (https://www.swpc.noaa.gov/products/wam-ipe). The WAM-IPE provides a specification of ionosphere conditions with forecasts two days in advance in response to solar, geomagnetic, and lower atmospheric forcing (Chou et al., 2023). It is different from IRI and Ne-Quick2 in that it is not an empirical model (climatology) but a space weather model.


The magnetic field is derived from the IGRF-13 (https://www.ngdc.noaa.gov/IAGA/vmod/igrf.html) (Alken et al., 2021). In essence, for each grid point of the electron density field the components of the magnetic field vector are computed and stored.

**2.4 Running the ray-tracing tool**

The software is written in Fortran and publicly available (Zus, 2025). After installation and compilation, the user must type the following in the command line of a Unix system to run the ray-tracing tool: *./prog yyyy-ddd-hh-mm station nwm*





*frequency source ionosphere* where *yyyy* denotes the four-digit year, *ddd* denotes the three-digit day of year, *hh* denotes the

two-digit hour, *mm* denotes the two-digit minute, *station* denotes the  list of stations, *nwm* denotes the numerical weather model, *frequency* denotes frequency (optical/microwave), *source* is the signal source (satellite/quasar) and *ionosphere* indicates if space-weather is switched on/off.

The numerical (space-) weather model data and the file for the station coordinates must be available in the *input* directory

and the solution is written in the *output* directory.

### 3 Experiments

#### 3.1 Benchmark comparison campaign


We validate the ray-tracing tool using data from the benchmark comparison campaign (Nafisi et al., 2012a). The participants of the campaign computed tropospheric delays for an elevation angle of 5° (every degree in azimuth) for one station in Germany (Wettzell) and one station in Japan (Tsukuba). We restrict the comparison to those participants utilizing the same weather model: KARAT (Hobiger et al., 2008), UNB (Urquhart et al., 2012) and VIENNA (Nafisi et al., 2012b).


Figure 1 shows the tropospheric delay as a function of the azimuth angle (elevation angle of 5°) for the station in Germany (Wettzell) as an example. Good agreement (differences on a cm level) is found among all the solutions. The differences are related to the transformation of the geopotential height to the geometric height and different interpolation approaches for the refractivity (Nafisi et al., 2012a). The approach utilized in our ray-tracing tool is similar to the approach utilized by UNB and

this explains the excellent agreement (differences on a mm level) for these two solutions. For new users of the ray tracing tool we recommend to run theses case studies first to check functionality. The possibility to run the two case studies from the benchmark comparison campaign is also interesting for the further development of the software as one can make changes to the code (e.g. optimization) and obtain quick feedback on the applicability of changes. However, the comparisons are restricted to two case studies. The comparison for dozens of stations locations and a long time period is subject to the next

section. This will spread light on both accuracy and speed.







**Figure 1: Tropospheric delays as a function of the azimuth angle (elevation angle of 5°) for the station Wettzell in Germany (1 January 2008, 0UTC). Different colours show different solutions: DNS (ray-tracing tool), KARAT (Hobiger et al., 2008), UNB (Urquhart et al., 2012) and VIENNA (Nafisi et al., 2012b).**

## 3.2 Comparison against RADIATE

We perform an one-to-one comparison of DNS with the open source software RADIATE (Landskron, 2018). We calculated tropospheric delays for the same stations, the same set of elevation and azimuth angles, the same NWM input data (ascii files are available from the VieVS ray-tracer https://vmf.geo.tuwien.ac.at/GRIB_TXT/), the same space geodetic technique (VLBI) and settings (e.g., Earth's radius of curvature and refractivity constants). We utilized the same PC (Intel(R) Core(TM) i7-9700 CPU @ 3.00GHz) and the same compiler (GNU Fortran compiler). Unlike RADIATE our ray-tracing tool utilizes OpenMP to speed up computation (the data throughput scales linearly with the number of cores). However, in the following comparisons, to be fair, a single core was utilized.





At first, we consider 2592 grid point coordinates with global coverage and a single epoch (15 February 2024, 12UTC). Figure 2 shows the differences in the tropospheric delays as a function of the elevation angle. In the zenith direction the differences can reach the mm level and at the elevation angle 3° the differences can reach the cm level. The mean and standard deviation as a function of the elevation angle indicate that the differences are random and not systematic for all elevation angles. Specifically, the standard deviation in the zenith is 0.5 mm and at the elevation angle of 3° it is 5.0 mm. The differences roughly scale with the elevation angle, i.e., 1/sin(e). Therefore Fig. 3 shows the relative differences in the tropospheric delays as a function of elevation angle. The relative differences are mainly below 0.1% for all elevation angles. These differences can be regarded small compared to the uncertainty in tropospheric delays caused by the uncertainty of the underlying refractivity field (also see discussion below). We hypothesize that the differences in the tropospheric delays are mainly caused by the way the undulation is applied and the way the interpolation (extrapolation) of refractivity is done. For example, RADIATE utilizes its own geoid height grid data file ('global_undulations_dint_lat1.000_dint_lon1.000.txt') whereas DNS utilizes the NIMA/GSFC WGS-84 15' geoid height grid data file ('ww15mgh.grd'). In addition, before the ray-tracing starts in RADIATE, the vertical resolution (height level resolution) from the weather model field is increased by interpolation and above the highest weather model level extrapolation using a standard atmosphere is carried out. In our ray-tracing tool the weather model field is used as is and above the highest weather model level the pressure, temperature and humidity are extrapolated. Specifically, we neglect the humidity, assume a constant temperature and calculate pressure from the hydrostatic equation. We support our hypothesize in that we performed another comparison where we utilize the geoid height grid data file from RADIATE in our ray-tracing tool and altered the extrapolation of refractivity in the RADIATE software accordingly. The respective altered routine is provided with our ray-tracing tool ( 'module_profilewise_refrHD_ECMWFmin.f90_fzus'). Specifically, the standard deviation in the zenith is reduced from 0.5 mm to 0.2 mm and at the elevation angle of 3° the standard deviation is reduced from 5.0 mm to 2.0 mm. We conclude that the quality of tropospheric delays from DNS is the same as provided by RADIATE. The advantage of DNS over RADIATE is the speed. For the same task, i.e., the computation of 120 times 2592 tropospheric delays, DNS turns out to be about 10 times faster than RADIATE (same PC, same compiler and a single core).

In order to explain the high speed of the ray-tracing tool it is necessary to dive into details of the underlying numerical algorithm. Two features allow the rapid and precise computation of the tropospheric delays. At first, the numerical solution of the differential equation is done utilizing an implicit finite difference scheme. Second, the numerical quadrature is done utilizing a higher order Newton's code formula (Simpson's rule). The combination of these two features allows us to use a low number of supporting points without losing the precision of the computed tropospheric delays (Zus et al., 2014). On the other hand, the solution by RADIATE is based on the piecewise application of Snell's law (Hofmeister et al., 2017). In other words, the numerical solution of the differential equation is done utilizing an explicit finite difference scheme and the



numerical quadrature is done utilizing a low order Newton's code formula (Trapezoidal rule). Therefore a large number of

supporting points must be utilizes to obtain high precision of the computed tropospheric delays.

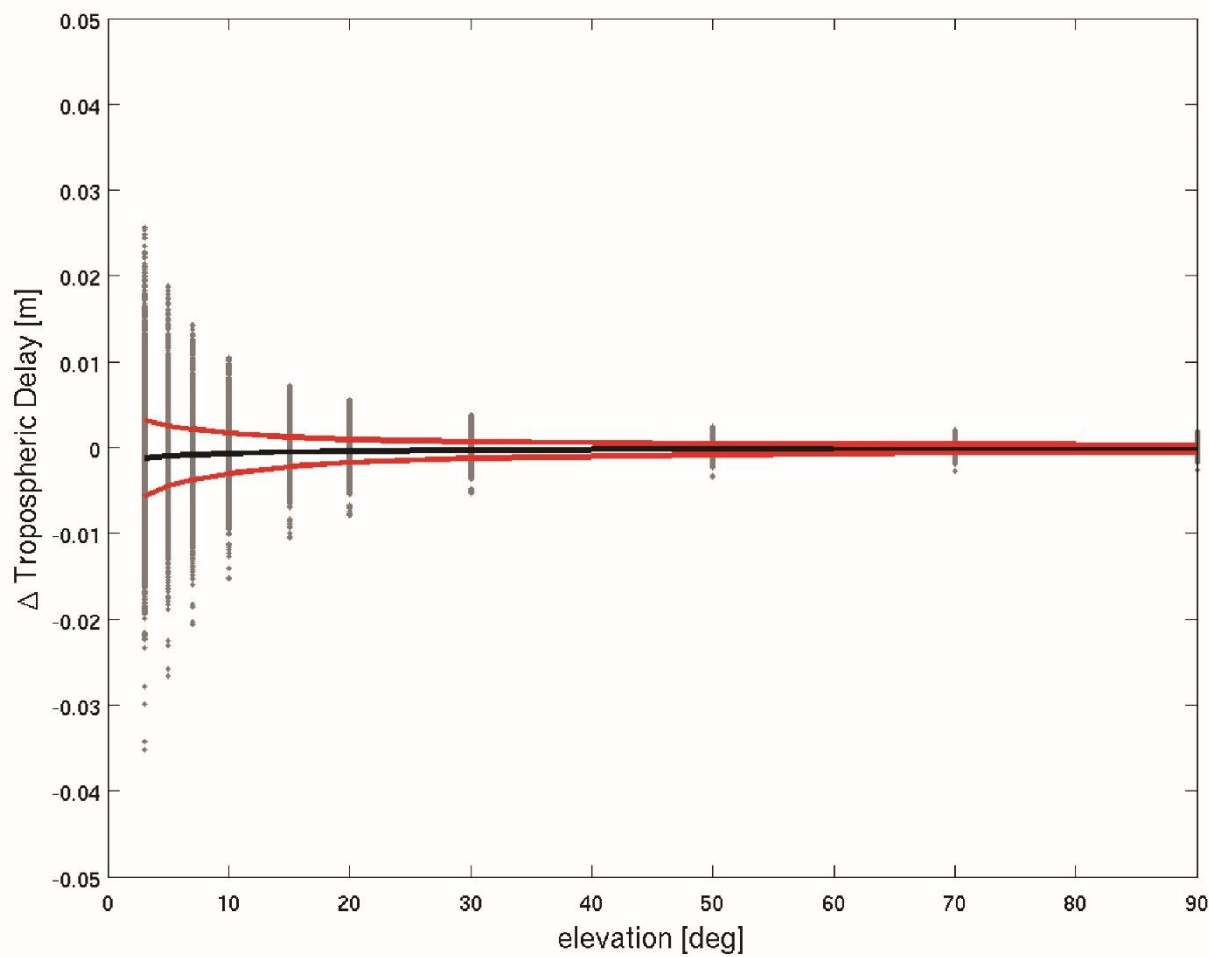

**Figure 2: Differences of tropospheric delays as a function of the elevation angle (15 February 2024, 12UTC). We consider 2592**
**globally distributed grid point coordinates and compute 120 tropospheric delays per grid point. The black line shows the mean**
**deviation and the red lines show the one-sigma deviation around the mean deviation.**



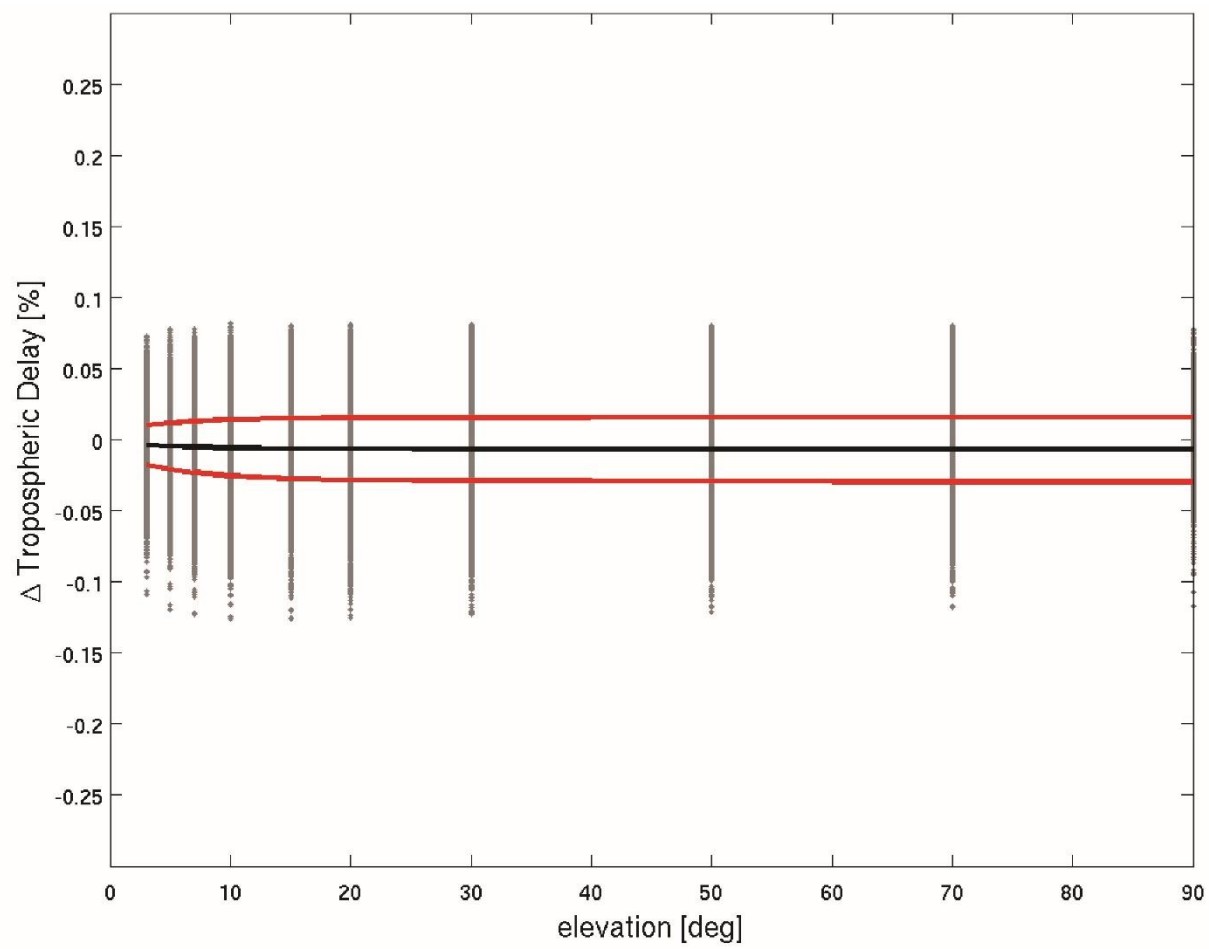

**Figure 3: Relative differences of tropospheric delays as a function of the elevation angle (15 February 2024, 12UTC). We consider**
**2592 globally distributed grid point coordinates and compute 120 tropospheric delays per grid point. The black line shows the**
**mean deviation and the red lines show the one-sigma deviation around the mean deviation.**

### 3.3 Application example: Comparison with GNSS tropospheric estimates

The raw measurements at a single GNSS station allow the estimation of the ZTD and tropospheric gradient. The ZTD contains information on the Integrated Water Vapor (IWV) at the station and, roughly spoken, the tropospheric gradient contains information on the horizontal IWV gradient at the station. This is why both GNSS tropospheric estimates are considered valuable in data assimilation for numerical weather prediction (Thundatil et al., 2024). Prior to data assimilation the comparison of GNSS and NWM tropospheric estimates is useful to better understand the information content and to estimate random and systematic deviations. The ray-tracing tool is well suited for this purpose. We are going to demonstrate this here utilizing a number of IGS core stations. For the time period 2018-2020 we compute every 6h tropospheric delays





with DNS and RADIATE. Based on the set of 120 tropospheric delays per station and epoch (the spacing in azimuth is 30° and the elevation angles are 3°, 5°, 7°, 10°, 15°, 20°, 30°, 50°, 70°, 90°) we calculate the Zenith Total Delay ZTD, the north gradient component $G_n$ and the east gradient component $G_e$ as follows


$$ZTD = T(90°, 0°) \tag{18}$$

$$G_n = \frac{\sum_{j=1}^{j=120} m_g(e_j)\sin(e_j)^2\cos(a_j)T(e_j,a_j)}{\sum_{j=1}^{j=k} m_g(e_j)^2\sin(e_j)^2\cos(a_j)^2}$$

$$G_e = \frac{\sum_{j=1}^{j=120} m_g(e_j)\sin(e_j)^2\sin(a_j)T(e_j,a_j)}{\sum_{j=1}^{j=k} m_g(e_j)^2\sin(e_j)^2\sin(a_j)^2}$$

where $m_g$ denotes the gradient mapping function and the indices $j$ indicate the specific azimuth and elevation angle. The formula for the north- and east gradient component is the result of a weighted least square adjustment. The idea behind this weighted least square adjustment is to mimic the way tropospheric gradients are estimated in the GNSS data analysis (Zus et

al., 2021). When we separate the tropospheric delay into the hydrostatic and wet delay we can separate the tropospheric gradient into two contributions which we call the hydrostatic and wet gradient component respectively.

As an example we select the station Wettzell (Germany). The standard deviation between ZTDs and gradient components derived from DNS and RADIATE, both utilizing ECMWF's operational analysis (IFS), is 0.6 mm and 0.01 mm,

respectively. We find no systematic deviation for the gradient components but a small systematic deviation of 0.1 mm for the zenith delays. Figure 4 shows the scatter of band-pass-filtered atmospheric delay coefficient differences between DNS and RADIATE. Across the frequency spectrum, the differences in the wet component are insignificant. The differences in the hydrostatic component are larger than in the wet component. We attribute this to the different way of extrapolating the refractivity above the model top (see discussion in the previous section).






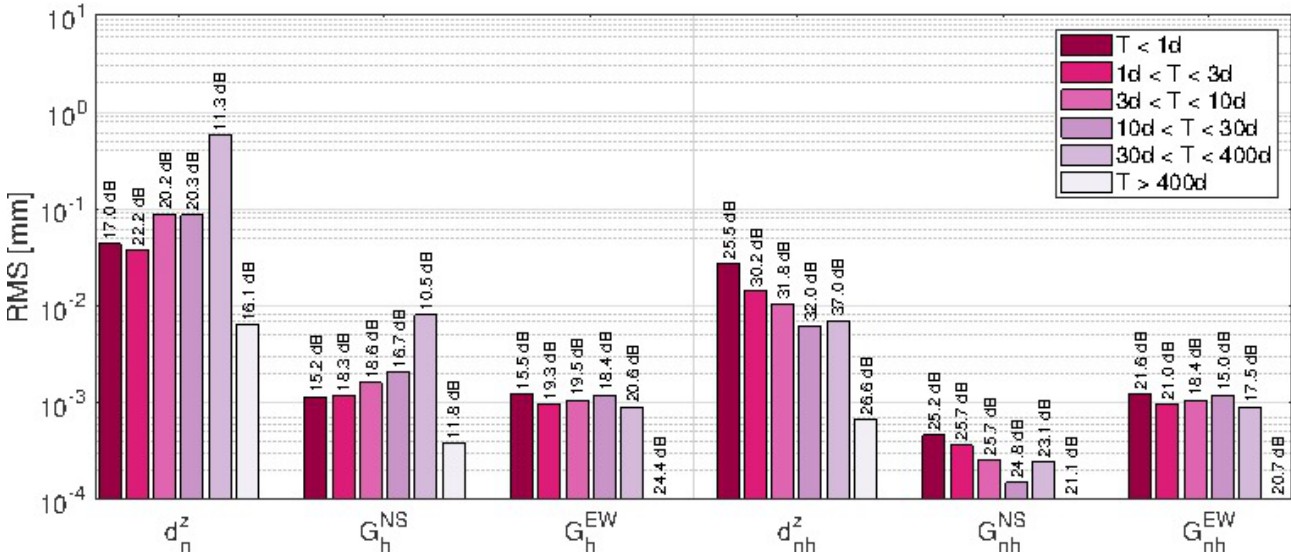

**Figure 4: Scatter of band-pass-filtered atmospheric delay coefficient differences between DNS and RADIATE employing IFS for the station Wettzell (Germany).**

Next we calculate ZTDs and gradient components utilizing the atmospheric reanalysis ERA5. This is possible with DNS but not with RADIATE. The standard deviation between IFS and ERA5 for the ZTD is 5.8 mm and for the gradient components it is between 0.2 and 0.3 mm. There is a systematic deviation of 1.6 mm in the zenith delays. Figure 5 shows the scatter of band-pass-filtered atmospheric delay coefficient differences between IFS and ERA5 for the station Wettzell (Germany). The differences induced by the different NWM for all parameters and frequency bands are about an order of magnitude larger

compared to the differences induced by employing the same NWM with different ray-tracing codes. As expected, the level of disagreement increases with increasing frequency.





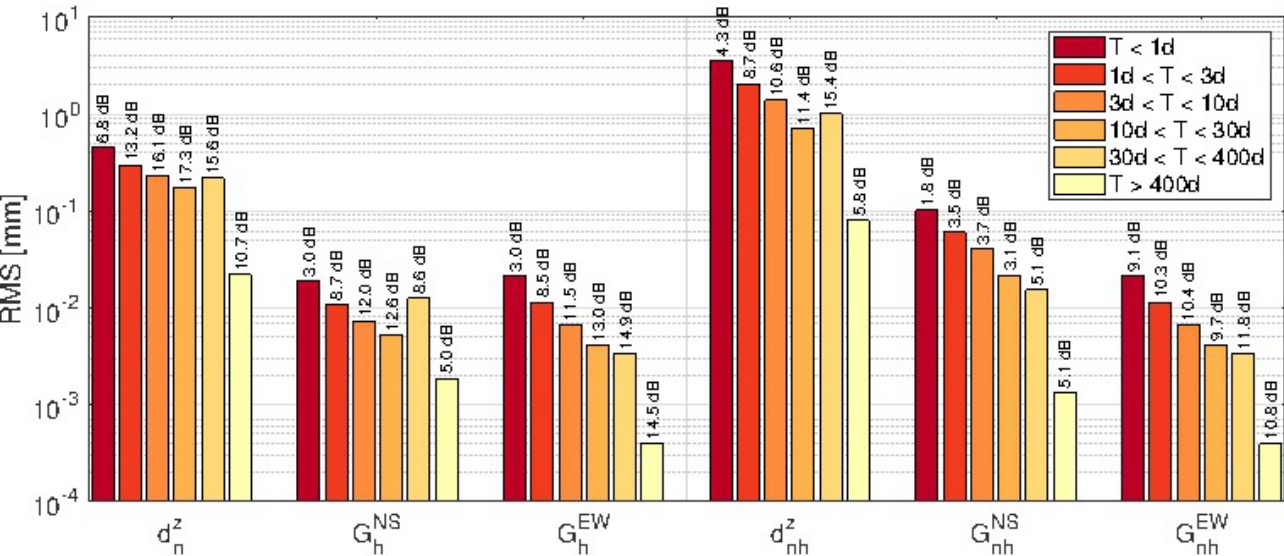

**Figure 5: Scatter of band-pass-filtered atmospheric delay coefficient differences between IFS and ERA5 employing DNS for the**
**station Wettzell (Germany).**

Finally, we compare the ZTDs and gradient components we calculated from ray-tracing (DNS and RADIATE) through
NWM fields (IFS and ERA5) employing DNS and RADIATE to GNSS tropospheric estimates. The GNSS tropospheric
estimates are taken from the contribution of TU Graz (Strasser et al., 2018) to the IGS Repro3 campaign. Figure 6 illustrates
the scatter of the band-pass-filtered differences between the ZTDs and gradients from ray-tracing and GNSS processing. As
expected, there is a tendency for better agreement between NWM-derived and GNSS-derived quantities with decreasing
frequency. For the selected station the results do not suggest a significantly better match between any of the three scenarios
to the GNSS data. We note that the GNSS-NWM discrepancies are about one order of magnitude larger compared to the
IFS-ERA5 discrepancies and about two orders of magnitude larger compared to the DNS-RADIATE differences.





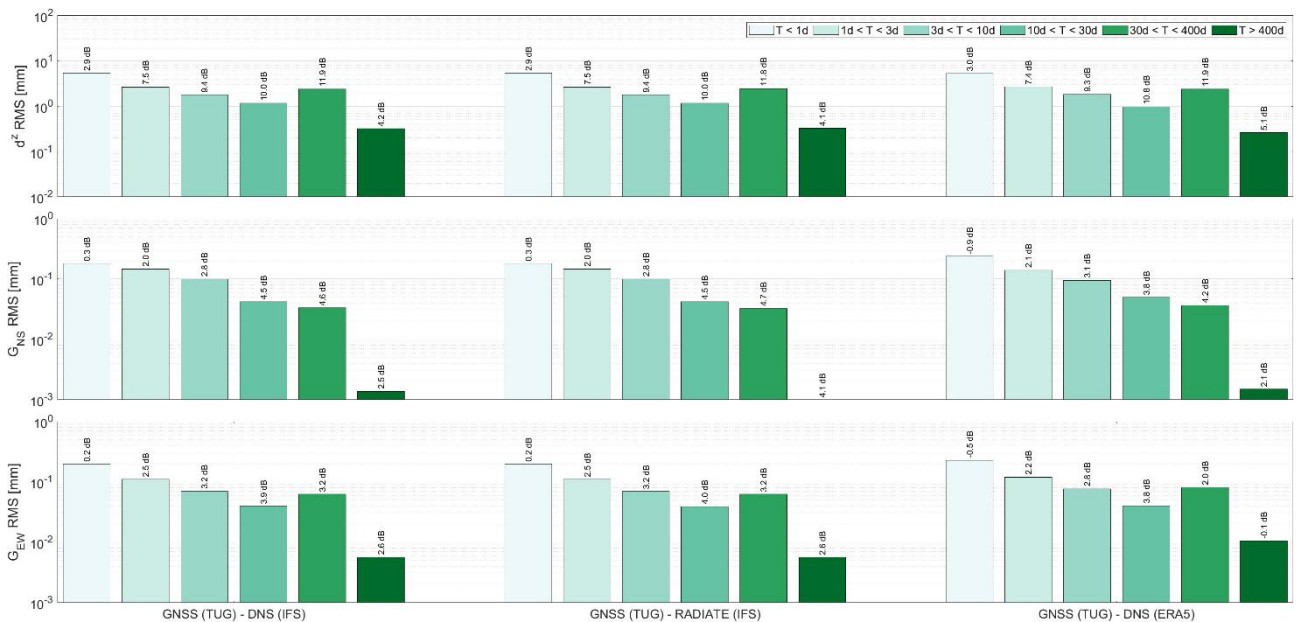

**Figure 6: Scatter of band-pass-filtered atmospheric delay coefficient differences between different solutions. GNSS (TUG) stands for the GNSS solution provided by TU Graz, DNS and RADIATE indicate the ray-tracing tool and IFS and ERA5 stand for the**
**two different weather models. All solutions are obtained for the station Wettzell (Germany).**

## 3.5 Comparison of ionospheric ray-path bending corrections

In the absence of another open source ray tracing tool capable of calculating ionospheric ray path bending corrections in a
realistic electron density field a one-to-one comparison is not possible. Therefore we make use of empirical formulae published in the literature. For example, under the assumption of a spherical layered ionosphere where the electron density profile is given by a single layer Chapman profile, Hoque et al., 2008 derived empirical formulas depending on the corresponding maximum ionization NmF2, the height of the maximum ionization hmF2, and the ionospheric scale height HF. Later Hoque et al., 2012 proposed new and simplified approaches for ionospheric ray path bending corrections and those
are the ones we are going to utilize here. In essence, in our ray-tracing tool we will utilize a single layer Chapman profile with NmF2 = $4.96 \cdot 10^{12}$ m$^{-3}$, hmF2 = 400 km and HF = 70 km. This specific single layer Chapman profile was utilized as an example by Hoque et al 2008. The VTEC is 143 TEC units (TECU). Figure 7 shows the ionospheric ray-path bending corrections as a function of the elevation angle from our ray-tracing tool and the empirical formula. The ionospheric ray-path bending corrections range from 0 mm in the zenith to about 20 mm at an elevation angle of 3°. The agreement between the
two solutions is reasonable. The differences are attributed to the inaccuracy of the empirical formula. We show this in that we added another independent numerical solution. This numerical solution is based on an implementation in MATLAB (it



works for single Chapman profiles only) where the ray-path is computed utilizing the ready to use bvp4c routine (Shampine et al., 2004). The bvp4c routine is a sophisticated (fourth-order) collocation method to solve BVPs. Indeed, the solution obtained from the bvp4c routine, which we regard as the most accurate solution, is in excellent agreement (differences are on

a sub-mm level) with the solution from our ray-tracing tool.

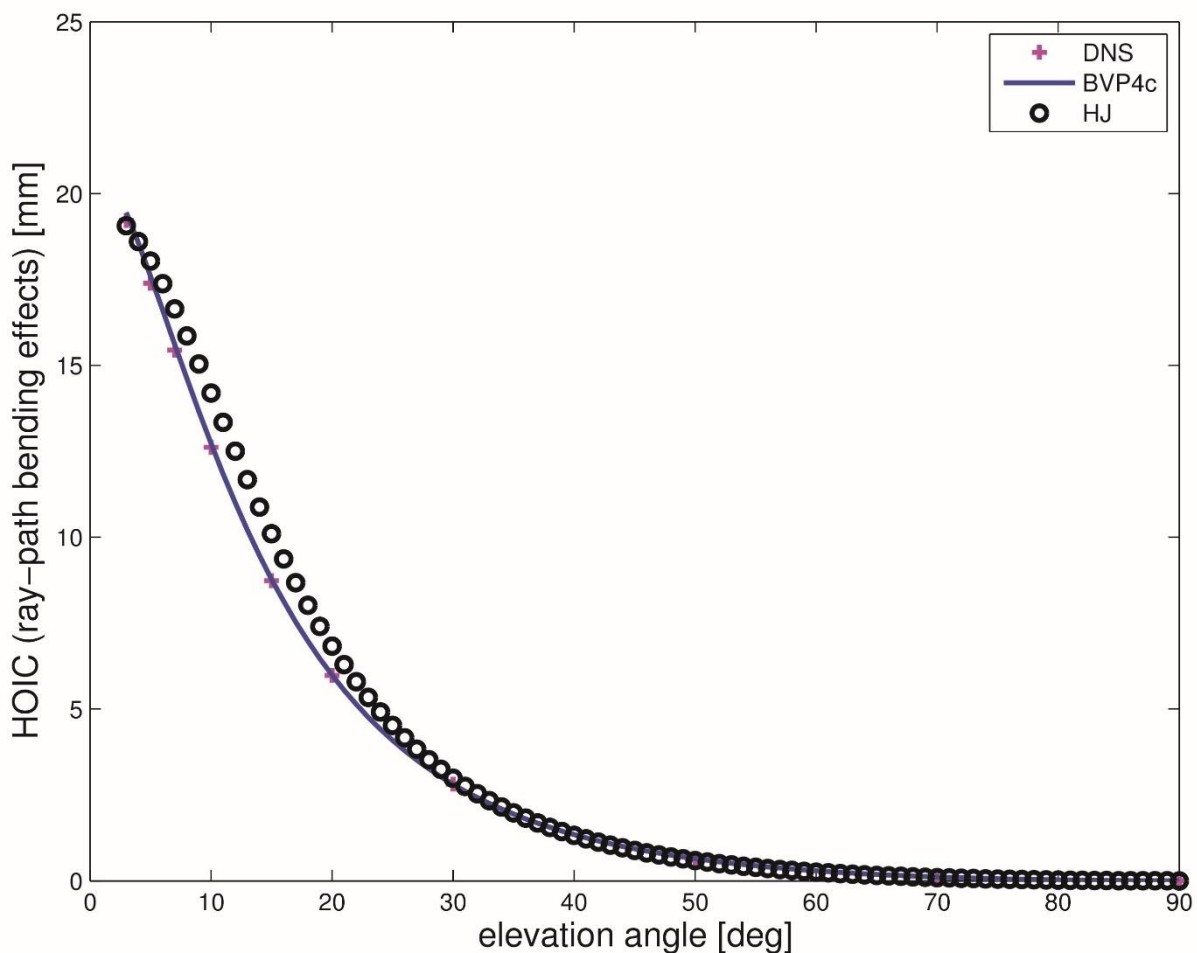

**Figure 7: Ionospheric ray-path bending effect as a function of the elevation angle. The underlying electron density profile equals a single layer Chapman profile where the maximum ionization NmF2 = 4.96 · $10^{12}$ m$^{-3}$, the height of the maximum ionization hmF2 =**

**400 km and the ionospheric scale height HF = 70 km (VTEC ~ 143 TECU). Different colours indicate different solutions: DNS (ray-tracing tool), HJ (Hoque et al., 2012), BVP4c (MATLAB, Shampine et al., 2004).**

## 3.6 Comparison with NeQuick 2 VTEC





We consider the same 2592 grid point coordinates with global coverage and a single epoch (15 March 2013, 12UTC) which is regarded representative for a period of high solar activity. The underlying electron density field comes from Ne-Quick2. The VTEC map is shown in Fig. 8. We can see the typical VTEC enhancement around the geomagnetic equator at noon. We verified that the VTEC calculated with the ray-tracing tool and the VTEC calculated with the source code of NeQuick2 model (https://t-ict4d.ictp.it/nequick2/source-code) yield differences well below 1 TECU for any grid point (not shown). The

differences can be explained by the fact that the ray-tracing tool requires interpolation of electron density in the gridded electron density field (prior to the ray-tracing we extract electron density profiles and assemble the electron density profiles to a 3D electron density field) whereas the source code of NeQuick2 model calculates the electron density for some point directly.

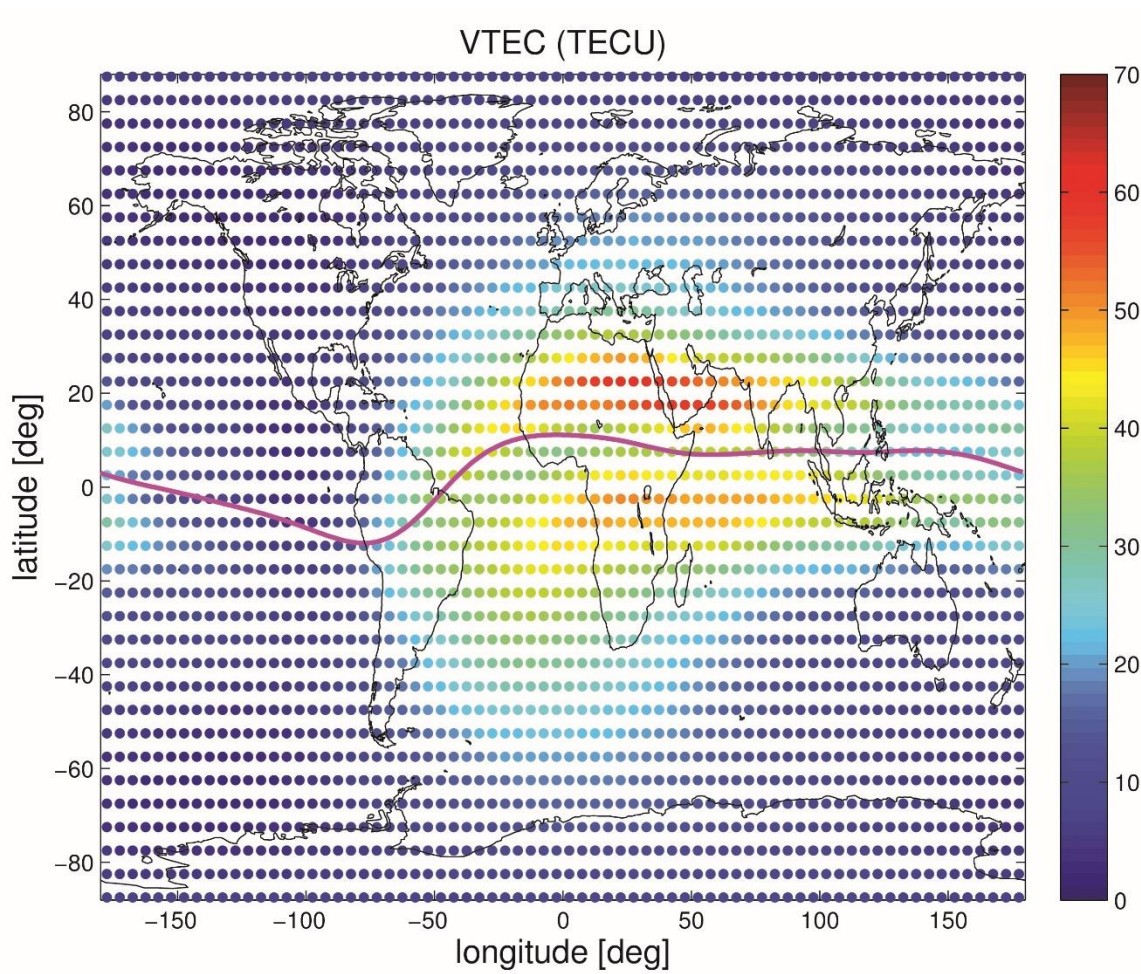


**Figure 8: The VTEC map calculated with the ray-tracing tool for one epoch (15 March 2015, 12UTC) utilizing the electron density field derived from Ne-Quick. The purple line indicates the geomagnetic equator.**


Next, for the same 2592 grid point coordinates with global coverage and the time period 1990-2019 (three solar cycles) we compute every 1h higher order ionospheric corrections. In order to save computer time and disk space each month consists of one day only (15th day of each month). We run the empirical model of the ionosphere with monthly mean values so that the day-to-day variability is small. For demonstration purposes it is sufficient to select one day per month. To start with we calculate the global mean VTEC and plot it as a function of the year in Fig. 9. The time evolution of the two global mean

VTEC values follow the solar cycle; large (small) during high (low) solar activity. How the higher-order ionospheric corrections, which are available for grid point coordinates with global coverage, can be utilized will be demonstrated in the next section.


**Figure 9: Global mean VTEC as a function of the year. Each month consists of one day only (15$^{\text{th}}$ day of each month) and each day consists of 24 epochs (1h resolution).**



### 3.7 Application example: Impact of higher order ionospheric corrections in precise point positioning

The calculation of higher-order ionospheric corrections using a realistic 3D electron density field was proposed by Kashcheyev et al., 2012. In fact, the underlying electron density field was derived from NeQuick2. They implemented a numerical homing-in ray-tracing algorithm to rigorously calculate satellite to station ray trajectories. The homing-in technique consists of the selection of the initial pulse (elevation angle and azimuth) at the satellite in a way the ray arrives exactly at the station. They implemented this by means of a dichotomizing search to adjust the initial azimuth and elevation angle. Using the homing-in ray-tracing algorithm for the two frequencies and the same satellite and station coordinates, exact ray trajectories and residual range errors were determined. The numerical simulations performed showed that higher-order ionospheric residual range errors may reach several cm (up to 5 cm) at low and mid- latitudes. Kashcheyev et al., 2012 argued that due to the computational complexity they investigated only two meridian cross-sections and they suggested to do further steps to analyse the space distribution of the corrections over the whole globe. The efficiency of our ray-tracing tool allows us to analyse the space distribution of the corrections over the whole globe (and several solar cycles). The key is that the differential equation is solved utilizing an implicit finite difference scheme (satellite and station coordinates are automatically part of the solution). However, in the following we will not analyse the higher-order ionospheric corrections themselves. We go one step further, probably more interesting, and show how higher-order ionospheric corrections leak (map) into the estimated parameters in the analysis of space geodetic data. We simulate Precise Point Positioning (PPP) with the GNSS. We performed a similar exercise in the past but instead of a short time period (one month) we explore the time period 1990-2019 (three solar cycles). We utilize the linearized observation equation where carrier-phase ambiguities are ignored (Zus et al., 2017b)

$$\Delta I(e,a) = -\boldsymbol{u}(e,a) \cdot \Delta\boldsymbol{r} + c \cdot \Delta t + m_w(e) \cdot \Delta ZTD + m_g(e) \cdot [\cos(a) \cdot \Delta G_n + \sin(a) \cdot \Delta G_e] \tag{19}$$

Here $\boldsymbol{u}$ denotes the tangent-unit vector of the station-satellite link, $\Delta\boldsymbol{r}$ denotes the coordinate residual vector, $\Delta t$ denotes the clock residual, $c$ denotes the vacuum speed of light and $m_w$ denotes the wet mapping function. The higher-order ionospheric corrections are determined for a realistic observation geometry with a cut-off elevation angle of 7°. We stack the linearized observation equations and obtain by the least-square adjustment the coordinate residual on a daily basis and the clock- and tropospheric parameter residuals epoch wise. The standard elevation angle dependent down weighting is applied in the least-square fit. For details the reader is referred to Zus et al., 2017b. We calculated ionospheric corrections for grid point coordinates with global coverage and stored the coefficient of the polynomial expansion. Hence, they can be evaluated at any station location and for any azimuth and elevation angle. We computed parameters residuals for more than 780 globally distributed stations (Zus et al., 2017b). The impact of higher order corrections depends on the station location (mainly the





modified dip latitude) and time. In particular the parameter residuals for stations around the geomagnetic equator ( +/- 40° around the geomagnetic equator) during high solar activity are affected. The most significant effects are the systematic ones and they can be summarized as follows. The stations appear to move southwards by up to 5 mm and this is consistent with

505 Kedar et al., 2003. The tropospheric north gradient components are systematically affected by up to 0.7 mm and this is consistent with Zus et al., 2017b. The zenith delays are systematically affected by up to 3 mm and this is consistent with Petri et al., 2010. As an example we chose one station in China (Wuhan) and plot the time series for the coordinate- and clock residual and the tropospheric parameters residuals in Fig. 10 and Fig. 11 respectively.

510

**Figure 10: Impact of higher-order ionospheric corrections on the estimated station coordinate and clock (expressed in mm) in PPP for one station in China (Wuhan) (Year 1990-2019). Each month consists of one day only (15th day of each month). Each day consists of 24 epochs (1h resolution).**





**Figure 11: Impact of higher-order ionospheric corrections on estimated zenith delays and tropospheric gradient components in PPP for one station in China (Wuhan) (Year 1990-2019). Each month consists of one day only (15th day of each month). Each day consists of 24 epochs (1h resolution).**

The time series shows that the impact mainly depends on the year. The dependency from the year stems from the solar activity. The high frequency variation in the clock and tropospheric residuals is due to the strong diurnal cycle; high (low) impact around noon (mid-night). Roughly spoken the impact in the parameter residuals follows the global mean VTEC (see Fig. 9). For this particular station in China (Wuhan) the impact in the x-coordinate reaches 0.2 mm, in the y-coordinate it reaches about 5 mm and in the up-component it reaches 0.5 mm. The station clock is 'effective' in absorbing the higher-order ionospheric correction as the impact in the clock residual reaches almost 3 mm. The impact in the east gradient component reaches 0.15 mm, in the north gradient component it reaches about 0.7 mm and in the zenith delay it reaches 2 mm. The time series show the known significant systematic impact on the estimated station latitude and the estimated north-gradient component. In addition, the time series reveals the significant systematic effect on the estimated zenith delay. It is





important to note that this significant systematic effect on the estimated zenith delay is mainly caused by the ray path bending effects ($\beta$ and $\gamma$ terms) and not by the higher order term in the ionospheric refractive index formula ($\alpha$ term). Therefore, we recommend the use of higher order ionospheric corrections derived from ray tracing. The limiting factor is the accuracy of the underlying electron density field. The empirical models running with monthly mean solar indices are likely to underestimate the effect. For this reason, and because we are also interested in real-time applications, we started to experiment with space weather model data (e.g. WAM-IPE) in the raytracing tool.

## 4 Conclusion

We present the open source ray tracing tool DNS that calculates atmospheric delay corrections for microwave and optical space geodetic observing systems. The atmospheric delay corrections are calculated between a ground station (a point close to the Earth's surface) and a satellite.

Comparing tropospheric delays from DNS with tropospheric delays from the popular RADIATE package, we find negligible differences when the refractivity fields and interpolation algorithms used are consistent: differences in zenith wet delays and gradients are typically less than 0.1 mm and 0.001 mm, respectively, across the frequency spectrum. Compared to changing the background model used to construct the refractivity fields from IFS to ERA5, the differences between DNS and RADIATE are at least an order of magnitude smaller. The advantages of DNS over RADIATE boil down to performance and flexibility. Even using a single core, DNS is multiple times (an order of magnitude) faster than RADIATE.

The ray tracing tool allows the study of signal propagation in the ionosphere. By comparing the VTEC output of the open source ray tracing tool with the VTEC output of the NeQuick2 model source code, we conclude that the two approaches are interchangeable in this respect. If it was just a matter of calculating VTEC then the logical choice is still the NeQuick2 model source code because the name says it all. However, the ray tracing tool also allows the 'quick' calculation of higher order ionospheric corrections. We compare the calculated higher order corrections with those from empirical formulas and find good agreement. As an application example, we show the impact of higher order ionospheric corrections on the estimated parameters in the analysis of space geodetic data. Such a simulation covering almost three solar cycles is not available yet. The efficient ray tracing tool makes this possible.

In conclusion, the ray tracing tool is expected to be useful for research and operational applications. We hope to stimulate and support further studies, such as the derivation of tropospheric (ionospheric) mapping functions from high-resolution numerical (space) weather models. DNS allows parallelization which allows it to exploit the parallel architecture of modern PCs and high-performance computing clusters. This feature of the software will support such studies.



**Code and data availability**


The source code of DNS and data from the benchmark comparison campaign is available at Zus (2025). The source code of RADIATE is available at Landskron (2018).


**Author contribution**

The original draft of the study was written by FZ and KB. FZ, KB, AHD and RT conducted the formal analysis and experiments. JW and GD reviewed and edited the paper.


**Competing interests**

The authors declare that they have no conflict of interest.

**Acknowledgments**

We acknowledge GNSS data provided by six of the analysis centers that contributed to the IGS repro 3: Center for Orbit Determination in Europe (COD), European Space Agency/ESOC (ESA), GeoForschungsZentrum (GFZ), Space geodesy team of the CNES (GRG), Jet Propulsion Laboratory (JPL), and Technische Universität Graz (TUG).

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
