# Peer review of "DNS (v1.0): An open source ray-tracing tool for space geodetic techniques"

_Geoscientific Model Development, 2024_

## Referee Comment (RC2)

Attachment to review on:

"DNS (v1.0): An open source ray-tracing tool for space geodetic techniques"

The authors present a ray tracing tool along with a description and validation of its functionality. The manuscript points to new opportunities that can be approached with this tool (high-precision modeling with global coverage). It is certainly an important achievement of interest for the community. Some points should be considered before publication, see here below.

**Comments:**

- (1) The title seems to be incomplete. Either there, or latest in the abstract, the meaning of "DNS" should be resolved. Furthermore, the authors stress in the manuscript that the tool allows quicker computations than before and that the number of computations can be increased up to an "application on global scale". This aspect could be reflected in the title and abstract.
- (2) The analysis in sections 3.6 and 3.7 leaves an open question. How are the parameters, presented in Fig. 9, 10, 11, affected on the sub-daily scale? Ionospheric corrections are computed with 1h resolution, however, the plots do not resolve the sub-daily scale. Is there an impact of the daily cycle (local solar incidence) similar to the impact of the solar cycle?
- (3) The state-of-the-art considerations to higher-order ionospheric corrections at the beginning of section 3.7 (line 470-480) are part of the motivation and should appear earlier in the introduction to explain early the goals of the study/tool.
- (4) The word "we" in the manuscript is not precisely used, see especially section 3.7. It refers to the author team of this manuscript, however, it also refers in some places to authors of previous papers!

Minor Comments:

- (a) Line 35: "... tropospheric delay is plugged into the observation equation ...", where can the reader find the observation equation?
- (b) Line 95: "... defined by the point P, the centre of the osculating sphere ...", better to clarify: "... defined by transmitter in point P, the centre of ... and the receiver in point Q".
- (c) Line 125: "... the ZHD from the empirical formula ...", which empirical formula, where to find it?
- (d) Eq. (8): here, \$N\$ is again the refractivity of the neutral gas? Please clarify.
- (e) Line 111 & 183 & 357: clarify why 120 corrections or delays are computed, 6 azimuth

bins x 10 elevation bins x ??

- (f) Line 195: please, explain the meaning of ds = dg.
- (g) Line 219: ECMWF data are defined on "137 pressure levels".
- (h) Line 263: "... ionosphere indicates if space-weather is switched on/off", does it mean that a background ionospheric climatology is always switched on and only space-weather effects can be switched?
- (i) Line 28: correct typo "... to run these cases ..."
- (j) Captions of Fig. 2 & 3: clarify that "differences between RADIATE and DNS" are shown.
- (k) Line 363 & 365: the gradients involve azimuth \$a\$ and elevation \$e\$, then different indices \$i\$, \$j\$ should be used, respectively, to run through all bin combinations!?
- (I) Line 373: sentence is unclear, maybe better: "standard deviation of ZTDs and gradient components considering the difference of DNS and RADIATE results ..."

(m)Captions of Fig. 4, 5 & 6: maybe clearer: "Root mean square (scatter) of ..."

- (n) Line 428: resolve the meaning of "... BVPs.".
- (o) Line 440: clarify origin of "... same 2592 grid point coordinates ..." lat bins x lon bins!?
- (p) Line 473: rather "There, ..." than "In fact, ..."
- (q) Line 497: provide a short explanation of "standard elevation angle dependent down weighting"
- (r) Line 552: maybe less casual: "... the NeQuick2 model source code because it provides this solution quickly."

---

## Author Comment (AC4)

*Point2point response:*

*We thank the reviewer for the positive feedback. Please, find the point2point response below (red and italic).*

The authors Florian Zus et al. introduce an open source ray-tracing tool for space geodetic techniques, named DNS (v1.0). This tool is very efficient in calculating ray-traced delays at optical and microwave frequencies in the neutral atmosphere and ionosphere, applying various numerical data sets such as operational or re-analysis datasets of the ECMWF and Nequick2.

With its speed (at least an order of magnitude faster than existing packages), it provides users with the possibility to improve parameterized models, such as mapping functions. Congratulations and thanks for providing this tool to the community.

Minor comments:

l122: Dealy -> Delay

*Will be corrected.*

l144: \theta is not used in equation (\varphi)

*Will be corrected.*

l179: angel -> angle

*Will be corrected.*

l319: hypothesis

*Will be corrected.*

l336: utilized

*Will be corrected.*

Figures 4 and 5: Should it read d_h^z instead of d_n^z ?

*You are right, it should be d_h^z. Will be corrected.*

Figure 6 is hardly readable

*The quality of figure 6 (4 and 5) will be improved in the revised version of the manuscript.*

---

## Author Comment (AC5)

**Point2point response:**

**We thank the reviewer for his questions and suggestions. Please, find the point2point response below (red and italic).**

Attachment to review on:

"DNS (v1.0): An open source ray-tracing tool for space geodetic techniques" The authors present a ray tracing tool along with a description and validation of its functionality. The manuscript points to new opportunities that can be approached with this tool (high-precision modeling with global coverage). It is certainly an important achievement of interest for the community. Some points should be considered before publication, see here below.

**Comments:**

(1) The title seems to be incomplete. Either there, or latest in the abstract, the meaning of "DNS" should be resolved. Furthermore, the authors stress in the manuscript that the tool allows quicker computations than before and that the number of computations can be increased up to an "application on global scale". This aspect could be reflected in the title and abstract.

**We thank the reviewer for this suggestion. We will explain the meaning of 'DNS' in the first few lines of the abstract.**

(2) The analysis in sections 3.6 and 3.7 leaves an open question. How are the parameters, presented in Fig. 9, 10, 11, affected on the sub-daily scale? Ionospheric corrections are computed with 1h resolution, however, the plots do not resolve the sub-daily scale. Is there an impact of the daily cycle (local solar incidence) similar to the impact of the solar cycle?

Thank you for this question. Fig. 9 is not affected on sub-daily scale as it shows the mean TEC (an average over the globe). However, the reviewer is right regarding Fig. 10 and 11. All parameters that are estimated epoch wise (on an hourly basis in our case), i.e. the zenith delay, the gradient components and the station clock show a strong diurnal cycle. Roughly speaking, the impact follows the local time; the impact at noon is large whereas the impact at midnight is small. The station coordinates are not affected as they are estimated on a daily basis (one estimate per day). In the revised version of the manuscript we will provide more details on that issue.

(3) The state-of-the-art considerations to higher-order ionospheric corrections at the

beginning of section 3.7 (line 470-480) are part of the motivation and should appear

earlier in the introduction to explain early the goals of the study/tool.

As suggested we will move part of this section in the introduction to explain early the goals of the tool.

(4) The word "we" in the manuscript is not precisely used, see especially section 3.7. It

refers to the author team of this manuscript, however, it also refers in some places to

authors of previous papers!

In the revised version of the manuscript we make clear who is meant by "we".

Minor Comments:

(a) Line 35: "... tropospheric delay is plugged into the observation equation ...", where

can the reader find the observation equation?

In the revised version of the manuscript we will refer to the (linearized) observation equation we make use of in the manuscript in section 3.7.

(b) Line 95: "... defined by the point P, the centre of the osculating sphere ...", better to

clarify: "... defined by transmitter in point P, the centre of ... and the receiver in point

Q".

The point P is the receiver and the point Q is the transmitter. Earth's radius is 'determined' at the point P. We will clarify this in the revised version of the manuscript.

(c) Line 125: "... the ZHD from the empirical formula ...", which empirical formula, where

to find it?

In the next sentence we provide the references for the empirical formulas (microwave and optical frequencies).

(d) Eq. (8): here, \$N\$ is again the refractivity of the neutral gas? Please clarify.

Yes. In the revised version of the manuscript we will clarify that N in Eq. 8 stands again for the refractivity of the neutral gas.

(e) Line 111 & 183 & 357: clarify why 120 corrections or delays are computed, 6 azimuth

bins x 10 elevation bins x ??

From our experience it is sufficient to compute 120 delays per station and epoch as they contain all information necessary to derive e.g. tropospheric mapping function coefficients. We will explain this in the revised version of the manuscript.

(f) Line 195: please, explain the meaning of ds = dg.

This means that the arc-length equals the geometric distance (no ray-path bending). In order to make this clear, we will state this in words in the revised version of the manuscript.

(g) Line 219: ECMWF data are defined on "137 pressure levels".

We utilize the ERA5 pressure level data and this is available with 37 pressure levels. We will clarify this at this point in the revised version of the manuscript.

(h) Line 263: "... ionosphere indicates if space-weather is switched on/off", does it mean

that a background ionospheric climatology is always switched on and only spaceweather effects can be switched?

No, it means that the user can switch on/off the ionosphere (space-weather model or climatology). In the revised version of the manuscript we will state this clearly.

(i) Line 28: correct typo "... to run these cases ..."

Will be corrected.

(j) Captions of Fig. 2 & 3: clarify that "differences between RADIATE and DNS" are

shown.

Will be corrected.

(k) Line 363 & 365: the gradients involve azimuth \$a\$ and elevation \$e\$, then different

indices \$i\$, \$j\$ should be used, respectively, to run through all bin combinations!?

The index j runs from 1 to 120. We want to avoid writing two sums, one with the index i=1,...,10 and another one with the index j=1,...,12.

(1) Line 373: sentence is unclear, maybe better: "standard deviation of ZTDs and gradient

components considering the difference of DNS and RADIATE results ..."

Will be corrected.

(m)Captions of Fig. 4, 5 & 6: maybe clearer: "Root mean square (scatter) of ..."

**Will be corrected.**

(n) Line 428: resolve the meaning of "... BVPs.".

BVP stands for Boundary Value Problem. It was introduced on page 4 at line 100.

(o) Line 440: clarify origin of "... same 2592 grid point coordinates ..." lat bins x lon bins!?

In the revised version of the manuscript we will explain that this means that the grid point coordinates are the same as those in the previous experiment (those introduced in section 3.2).

(p) Line 473: rather "There, ..." than "In fact, ..."

Will be corrected.

(q) Line 497: provide a short explanation of "standard elevation angle dependent down

weighting"

In the revised version of the manuscript we will explain that this means that the elevation angle dependent down-weighting is done utilizing  $1/\sin(e)$  where e denotes the elevation angle.

(r) Line 552: maybe less casual: "... the NeQuick2 model source code because it provides

this solution quickly."

Will be corrected as suggested.

---

## Author Response (AR1)

*Review#1:*

*We thank the reviewer for the positive feedback. Please, find the point2point response below (blue and italic). The line number corresponds to the marked up version of the manuscript.*

The authors Florian Zus et al. introduce an open source ray-tracing tool for space geodetic techniques, named DNS (v1.0). This tool is very efficient in calculating ray-traced delays at optical and microwave frequencies in the neutral atmosphere and ionosphere, applying various numerical data sets such as operational or re-analysis datasets of the ECMWF and Nequick2.

With its speed (at least an order of magnitude faster than existing packages), it provides users with the possibility to improve parameterized models, such as mapping functions. Congratulations and thanks for providing this tool to the community.

Minor comments:

l122: Dealy -> Delay

*Corrected, see L127.*

l144: \theta is not used in equation (\varphi)

*Corrected, see L150.*

l179: angel -> angle

*Corrected, see L184.*

l319: hypothesis

*Corrected, see L330.*

l336: utilized

*Corrected, see L347.*

Figures 4 and 5: Should it read d_h^z instead of d_n^z ?

*You are right, it should be d_h^z. Corrected, see Figure 4 and 5.*

Figure 6 is hardly readable

*The quality of figure 6 (4 and 5) is improved in the revised version of the manuscript.*

*Review#2:*

*We thank the reviewer for his questions and suggestions. Please, find the point2point response below (blue and italic). The line number corresponds to the marked up version of the manuscript.*

Attachment to review on:

„DNS (v1.0): An open source ray-tracing tool for space geodetic techniques"

The authors present a ray tracing tool along with a description and validation of its

functionality. The manuscript points to new opportunities that can be approached with this tool

(high-precision modeling with global coverage). It is certainly an important achievement of

interest for the community. Some points should be considered before publication, see here

below.

Comments:

(1) The title seems to be incomplete. Either there, or latest in the abstract, the meaning of

"DNS" should be resolved. Furthermore, the authors stress in the manuscript that the

tool allows quicker computations than before and that the number of computations can

be increased up to an "application on global scale". This aspect could be reflected in

the title and abstract.

*We thank the reviewer for this suggestion. We now explain the meaning of 'DNS' in the first few lines of the abstract, see L13.*

(2) The analysis in sections 3.6 and 3.7 leaves an open question. How are the

parameters, presented in Fig. 9, 10, 11, affected on the sub-daily scale? Ionospheric

corrections are computed with 1h resolution, however, the plots do not resolve the subdaily scale. Is there an impact of the daily cycle (local solar incidence) similar to the

impact of the solar cycle?

*Thank you for this question. Fig. 9 is not affected on sub-daily scale as it shows the mean TEC (an average over the globe). However, the reviewer is right regarding Fig. 10 and 11. All parameters that are estimated epoch wise (on an hourly basis in our case), i.e. the zenith delay, the gradient components and the station clock show a strong diurnal cycle. Roughly speaking, the impact follows the local time; the impact at noon is large whereas the impact at midnight is small. The station coordinates are not affected as they are estimated on a daily basis (one estimate per day). In the revised version of the manuscript we provide more details on that issue, see L535.*

(3) The state-of-the-art considerations to higher-order ionospheric corrections at the beginning of section 3.7 (line 470-480) are part of the motivation and should appear earlier in the introduction to explain early the goals of the study/tool.

*As suggested we moved part of this section in the introduction, see L63 and L483. On the other hand we think that details of the study by Kashcheyev et al., 2012 should stay in section 3.7.*

(4) The word "we" in the manuscript is not precisely used, see especially section 3.7. It refers to the author team of this manuscript, however, it also refers in some places to authors of previous papers!

*In the revised version of the manuscript we make clear who is meant by "we", see L498.*

Minor Comments:

(a) Line 35: "… tropospheric delay is plugged into the observation equation …", where can the reader find the observation equation?

*In the revised version of the manuscript we now provide a reference, see L40.*

(b) Line 95: "… defined by the point P, the centre of the osculating sphere …", better to clarify: "… defined by transmitter in point P, the centre of … and the receiver in point Q".

*The point P is the receiver and the point Q is the transmitter. Earth's radius is 'determined' at the point P. We clarify this in the revised version of the manuscript, see L99.*

(c) Line 125: "… the ZHD from the empirical formula …", which empirical formula, where to find it?

*In the next sentence we provide the references for the empirical formulas (microwave and optical frequencies).*

(d) Eq. (8): here, $N$ is again the refractivity of the neutral gas? Please clarify.

*Yes. In the revised version of the manuscript we clarify that N in Eq. 8 stands again for the refractivity of the neutral gas, see L149.*

(e) Line 111 & 183 & 357: clarify why 120 corrections or delays are computed, 6 azimuth bins x 10 elevation bins x ??

*From our experience it is sufficient to compute 120 delays per station and epoch as they contain all information necessary to derive e.g. tropospheric mapping function coefficients. We now explain this in the revised version of the manuscript, see L118.*

(f) Line 195: please, explain the meaning of $ds = dg$.

*This means that the arc-length equals the geometric distance (no ray-path bending). In order to make this clear, we state this in words in the revised version of the manuscript, see L200.*

(g) Line 219: ECMWF data are defined on "137 pressure levels".

*We utilize the ERA5 pressure level data and this is available with 37 pressure levels. We clarify this at this point in the revised version of the manuscript, see L225.*

(h) Line 263: "… ionosphere indicates if space-weather is switched on/off", does it mean

that a background ionospheric climatology is always switched on and only spaceweather effects can be switched?

*No, it means that the user can switch on/off the ionosphere (space-weather model or climatology). In the revised version of the manuscript we state this more, see L273.*

(i) Line 28: correct typo "… to run these cases …"

*Corrected, see L292 and L295.*

(j) Captions of Fig. 2 & 3: clarify that "differences between RADIATE and DNS" are

shown.

*Corrected, see caption of Fig. 2 and 3.*

(k) Line 363 & 365: the gradients involve azimuth $a$ and elevation $e$, then different

indices $i$, $j$ should be used, respectively, to run through all bin combinations!?

*The index j runs from 1 to 120. We want to avoid writing two sums, one with the index i=1,...,10 and another one with the index j=1,...,12.*

(l) Line 373: sentence is unclear, maybe better: "standard deviation of ZTDs and gradient

components considering the difference of DNS and RADIATE results …"

*Corrected, see L384.*

(m) Captions of Fig. 4, 5 & 6: maybe clearer: "Root mean square (scatter) of …"

*Corrected, see caption of Fig. 4,5 and 6.*

(n) Line 428: resolve the meaning of "… BVPs.".

*BVP stands for Boundary Value Problem. It was introduced on L105.*

(o) Line 440: clarify origin of "… same 2592 grid point coordinates …" lat bins x lon bins!?

*In the revised version of the manuscript we explain that this means that the grid point coordinates are the same as those in the previous experiment (section 3.2), see L451.*

(p) Line 473: rather "There, …" than "In fact, …"

*Corrected, see L484.*

(q) Line 497: provide a short explanation of "standard elevation angle dependent down weighting"

*In the revised version of the manuscript we explain that this means that the elevation angle dependent down-weighting is done utilizing 1/sin(e) where e denotes the elevation angle, see L508.*

(r) Line 552: maybe less casual: "… the NeQuick2 model source code because it provides this solution quickly."

*Corrected, see L569.*